# The Effects of Fucoidan Derived from *Sargassum filipendula* and *Fucus vesiculosus* on the Survival and Mineralisation of Osteogenic Progenitors

**DOI:** 10.3390/ijms25042085

**Published:** 2024-02-08

**Authors:** Dhanak Gupta, Diana C. Martinez, Miguel Angel Puertas-Mejía, Vanessa L. Hearnden, Gwendolen C. Reilly

**Affiliations:** 1Department of Materials Science and Engineering, University of Sheffield, Sheffield S1 3JD, UK; dinama2012@gmail.com (D.C.M.); g.reilly@sheffield.ac.uk (G.C.R.); 2INSIGNEO Institute for in Silico Medicine, University of Sheffield, Sheffield S1 3JD, UK; 3School of Dentistry, Institute of Clinical Sciences, College of Medical and Dental Sciences, University of Birmingham, 5 Mill Pool Way, Edgbaston, Birmingham B5 7EG, UK; 4Faculty of Material Science and Engineering, Warsaw University of Technology, Wołoska 141, 02-507 Warszawa, Poland; 5Facultad de Ciencias Exactas y Naturales, Universidad de Antioquia, UdeA, Calle 70 No. 52-21, Medellín 050010, Colombia

**Keywords:** osteosarcoma, human embryonic-derived mesenchymal progenitor cells, fucoidan, bone healing, apoptosis, necrosis, brown algae, mitochondria

## Abstract

Osteosarcoma is a bone cancer primarily affecting teenagers. It has a poor prognosis and diminished quality of life after treatment due to chemotherapy side effects, surgical complications and post-surgical osteoporosis risks. The sulphated polysaccharide fucoidan, derived from brown algae, has been a subject of interest for its potential anti-cancer properties and its impact on bone regeneration. This study explores the influence of crude, low-molecular-weight (LMW, 10–50 kDa), medium-molecular-weight (MMW, 50–100 kDa) and high-molecular-weight (HMW, >100 kDa) fractions from *Sargassum filipendula*, harvested from the Colombian sea coast, as well as crude fucoidan from *Fucus vesiculosus*, on a specific human osteoprogenitor cell type, human embryonic-derived mesenchymal stem cells. Fourier transform infrared spectroscopy coupled with attenuated total reflection (FTIR-ATR) results showed the highest sulphation levels and lowest uronic acid content in crude extract from *F. vesiculosus*. There was a dose-dependent drop in focal adhesion formation, proliferation and osteogenic differentiation of cells for all fucoidan types, but the least toxicity was observed for LMW and MMW. Transmission electron microscopy (TEM), JC-1 (5,50,6,60-tetrachloro-1,10,3,30-tetraethylbenzimi-dazolylcarbocyanine iodide) staining and cytochrome c analyses confirmed mitochondrial damage, swollen ER and upregulated autophagy due to fucoidans, with the highest severity in the case of *F. vesiculosus* fucoidan. Stress-induced apoptosis-like cell death by *F. vesiculosus* fucoidan and stress-induced necrosis-like cell death by *S. filipendula* fucoidans were also confirmed. LMW and MMW doses of <200 ng/mL were the least toxic and showed potential osteoinductivity. This research underscores the multifaceted impact of fucoidans on osteoprogenitor cells and highlights the delicate balance between potential therapeutic benefits and the challenges involved in using fucoidans for post-surgery treatments in patients with osteosarcoma.

## 1. Introduction

Osteosarcoma is a type of bone cancer that is most common in teenagers [1]. It has a poor prognosis, and the quality of life is reduced even after treatment. The most common route of therapy requires multidisciplinary management, including neoadjuvant multiagent chemotherapy and surgical resection, followed by post-operative chemotherapy. The surgical resection requires the removal of wide margins, i.e., the removal of a few centimetres of normal tissue or bone away from the level of tumor edges [2]. Chemotherapy is known to cause short- and long-term side effects, including cardiac failure, hearing loss, nephrotoxicity, gonadal dysfunction and early menopause [3]. Moreover, chemotherapy-induced tissue injury is not only limited to the cancer cells but also to healthy cells surrounding the tumor. In some cases, the sarcoma even becomes chemotherapy-resistant [4].

After surgical removal, reconstruction is usually via megaprosthesis (large-scale implants) or biological reconstructive surgeries using methods such as autografts, allografts, cadaveric grafts or allograft–prosthesis composites [2]. However, these are accompanied by disadvantages such as the mechanical failure of megaprosthesis due to wear and tear; limited availability of cadaveric bone and its possible association with infection, graft fracture, non-union and osteoarthritis; and autografting, which often involves a second surgery, causing further patient discomfort and donor site morbidity. These surgical procedures are also prolonged and repeated, which accounts for nearly 10% of the reported cases of periprosthetic infections [5]. Moreover, neither of the available bone replacement materials can simultaneously prevent or promote bone repair and inhibit further recurrence of osteosarcoma.

A higher risk of osteoporosis and bone fracture development is also now reported in patients with osteosarcoma post-surgery as long-term studies revealed lower bone mineral density measurements of non-operative sites in these patients, whether children [6] or adults [7]. Anti-resorptive drugs, known as bisphosphonates, are often prescribed to patients with osteosarcoma before surgery and at the time of chemotherapy for bone pain management. Unfortunately, bisphosphonates can cause gastrointestinal disturbances, oesophagitis, fever and bone and muscle pain, with a possibly increased risk of oesophageal cancer and jaw osteonecrosis associated with denosumab [8]. As a result, there is clearly an immediate need for the management of bone remodeling post-surgery in patients with osteosarcoma not only at the site of resection but also at the systemic level for holistic patient recovery and reduced implant failure.

Fucoidan is derived from brown algae and is a sulphated polysaccharide. It can vary greatly in its chemical structure and, therefore, in its chemical properties and bioactivity. This variation arises mainly due to the species and habitat of algae, harvest time, extraction process and the molecular weight of the fucoidan. It has been shown that fucoidans have several bioactive properties, such as anti-cancer, anti-inflammatory, anti-coagulant, anti-virus, anti-prion, anti-oxidant, anti-hypertensive and anti-fibrotic activities and osteoinductivity (reviewed in [9,10]). Also, fucoidan is commonly used in food supplements and as an adjuvant in the pharmaceutical and cosmetic industries [11,12].

Previously, some studies investigated fucoidans as anti-cancer drugs against osteosarcoma [13,14,15], and recently, we also demonstrated for the first time that fucoidans from *Fucus vesiculosus* and *Sargassum filipendula* have anti-cancer properties, which vary in relation to the penetration ability of these fucoidans into MG63 osteosarcoma-derived cells [16]. While fucoidan from *F. vesiculosus* caused cell death via stress-induced apoptosis, fucoidan from *S. filipendula* led to stress-induced necrosis-like cell death. We also showed that as the molecular weight of *S. filipendula*-derived fucoidan increased, the anti-cancer activity also increased [16].

An ideal bone replacement graft in patients with osteosarcoma would have a positive impact on the bone regeneration process in addition to the anti-cancer activities. Interestingly, some studies reported fucoidan to be bone regenerative [17,18,19,20,21,22,23,24,25,26,27]. Changotade et al. [17] first reported that fucoidan (10 µg/mL) from brown algae increased the number of human osteoblasts and their bone formation ability over 45 days. Human alveolar bone-marrow-derived mesenchymal stem cells also showed increased proliferation; alkaline phosphatase (ALP) activity; *Runx2*, *Col1ɑ1* and *OCN* gene expression; and mineralisation attributed to enhanced Jun N-terminal kinase (JNK) and extracellular signal-regulated kinase (ERK)-dependent signaling in response to bone morphogenic protein (BMP2)-Smad 1/5/8 signaling, in the presence of fucoidan from *Laminaria japonica* (61.5% polysaccharides and 23.5% sulphate) [20].

For a useful osteosarcoma therapy and better quality of patient life after treatment, a pharmacological agent would ideally have a selective action, causing less damage to the healthy cells than the cancer cells. In this regard, fucoidans from different sources are particularly attractive since they have been reported to have positive effects on healthy differentiating osteogenic cells and/or inhibit cancerous cells. Therefore, in this study, we investigated the effects of the same fucoidan doses known to damage MG63 (osteosarcoma cells) on a non-immortalised human osteoprogenitor cell type (human embryonic-derived mesenchymal progenitor cells, hES-MP). We explored the effect of fucoidans from *F. vesiculosus* and low, medium- and high-molecular-weight (LMW, MMW and HMW) fractions of fucoidan derived from *S. filipendula* on cell proliferation, cell death and osteogenesis for bone regeneration applications.

## 2. Results

### 2.1. Characterisation of Fucoidans

The FTIR-ATR analysis showed the characteristic absorption bands of sulphated polysaccharides O-C-O, S=O and C-O-S for all the fucoidans at 1603.3, 1218.6 and 823.8 cm^−1^, respectively (Figure 1). There was a broad band observed at 3400 cm^−1^ corresponding to the O-H bond extension band, a small peak around 2800–2900 cm^−1^ characteristic of the C-H stretching of the pyranoside ring and the C-6 group associated with fucose and glucose units in the polysaccharides. From 1200–950 cm^−1^, the C-C and C-O vibration bands of the pyranoside ring and the C-O-C vibration of the glycosidic bond are observed. The presence of high-intensity absorption bands is characteristic for the polysaccharides. Finally, the vibration band at 823.8 cm^−1^ is associated with the C-O-S bending band, which is typical of substitutions at the axial C-4 position. Table 1 shows the assignment of functional groups in the fucoidans based on FTIR-ATR analysis. There was significantly higher sulphation levels and glycosidic bonds and lower uronic acid content in the crude extract from *F. vesiculosus* compared to all other fucoidans.

### 2.2. Effect of Fucoidan on hES-MP Attachment

To assess the attachment of hES-MPs in the culture medium supplemented with fucoidan, cells were seeded in media containing a range of doses of different fucoidan types. The results (Figure 2A) showed that as the fucoidan dose increased, the measured hES-MP metabolic activity dropped. Comparison of the two crude extracts showed more severe reduction in metabolic activity in the case of *S. filipendula* at all doses. Additionally, comparison of the different molecular weight fractions of fucoidans from *S. filipendula* showed the most severe negative effects in the cases of MMW and HMW and they were less so in the case of LMW by the 100 μg/mL fucoidan dose, which is similar to what has been reported for MG63 cells.

Immunostaining for actin and vinculin (Figure 2B) showed that in control conditions hES-MPs tended to have a flat and feather-like features with vinculin expressed at the edges of the cells. In the presence of 0.5 μg/mL of any fucoidan type, hES-MPs seemed to retain this expression pattern for vinculin, while at the 100 μg/mL dose there was negligible vinculin observable at the edges of the cells. These results suggested that all fucoidans were inhibiting cell attachment in a dose-dependent manner with the worst effects in the case of both crude extracts and also MMW and HMW fucoidans from *S. filipendula*.

### 2.3. Effect of Fucoidan on hES-MP Proliferation and Morphology

In a preliminary investigation (Appendix A), cells were cultured in the presence of fucoidan for 5 days and then stained with Giemsa for a visual inspection. The results indicated that a low dose of 0.5 μg/mL did not seem to affect the cell morphology. At a higher dose of 10 μg/mL, the hES-MPs had a distorted cytoplasm and morphology in the presence of crude and HMW extracts of fucoidans from *S. filipendula* but not in the presence of other fucoidans. Finally, at the highest dose of 100 μg/mL, LMW fucoidan from *S. filipendula* seemed to be the least toxic, while all other fucoidans significantly distorted the cell morphology and led to the formation of cell debris. In our previous study, the MG63 cells had also incurred similar damage in the presence of these fucoidans, especially the crude and higher-molecular-weight fractions, after only 3 days of treatment [16].

As the osteogenic differentiation in vitro is usually investigated from 3–4 weeks, the cell culture period was extended up to 21 days in media supplemented with 0 to 10 μg/mL of different fucoidans. Since 100 μg/mL of fucoidans was toxic to the cells within the time frame of 5 days (Appendix A), this concentration was eliminated from this experiment. The results (Figure 3A) showed a consistent dose-dependent drop in the metabolic activity after treatment with all types of fucoidan, with the strongest effect for *F. vesiculosus* fucoidan at 10 μg/mL at all time points. Additionally, there was a molecular-weight-dependent effect of fucoidan from *S. filipendula* on cell metabolic activity. The highest metabolic activity was in the case of cells treated with the LMW fraction, followed by the MMW fraction and the least in the case of cells treated with the HMW fraction at all time points, especially at doses more than 1 μg/mL. A confirmatory DNA content measurement (Appendix A) on Day 21 of culture also showed a dose-dependent decrease in cell numbers after treatment with all fucoidan types.

Giemsa staining on Day 21 (Figure 3B) indicated that the cells in control conditions had clearly defined nuclei with elongated and flat morphology. However, the treated cells did not have the same integrity, alignment or cytoskeletal structure and had the appearance of dying cells. For HMW, this negative effect on cell morphology was visible even at lower doses of 0.5 and 1 μg/mL but, for other fucoidans, the toxic doses were higher than 1 μg/mL. These results clearly indicate that fucoidan inhibits cell growth and affects the cytoskeleton in a dose-dependent manner with the worst effect in the case of the HMW fraction.

### 2.4. Cell Cycle Analysis

Human ES-MP cell cycle analysis showed that after 5 days of treatment with a 10 μg/mL dose of fucoidan from *F. vesiculosus*, the sub-G1 and G1 populations increased by 6% and 4%, respectively, and the S and G2 populations decreased by 1% and 2%, respectively (Figure 4). This indicated an accumulation of cell debris after this fucoidan treatment. A similar effect was reported for the MG63 cells after this fucoidan treatment [16]. Treatments with 10 μg/mL of fucoidans from *S. filipendula*, however, did not lead to an increase in sub-G1 populations for hES-MPs but there was 1–5% rise in G1 populations for all of these fucoidans, which may indicate a G1 phase arrest. Collectively, these results suggested that fucoidans from different brown seaweed species may have acted differently on the hES-MPs, similar to as seen for the MG63 cells previously [16].

### 2.5. Cell Death Analysis

Annexin V/Propodium iodide (PI) staining was also performed. The results (Figure 5) showed that 0.5 and 100 μg/mL of *F. vesiculosus* fucoidan led to significantly reduced viable cells, more early apoptotic cells and more dead cells compared to the respective vehicle control (0 μg/mL). Moreover, there was slightly reduced percentage of late apoptotic (Annexin V+ and PI+) cells and the presence of a split population (unstained) in the viable gated region with the 100 μg/mL treatment with fucoidan from *F. vesiculosus*, which was less apparent for other fucoidans at the same dose. To confirm these observations, the cells were seeded at a higher cell density and treated with 0, 0.5 and 100 μg/mL of fucoidan from *F. vesiculosus* for 10 days. The results (Appendix A) confirmed that crude fucoidan from *F. vesiculosus* led to reduced viability, enhanced apoptosis and a pronounced presence of unstained cell debris. A similar pattern of Annexin V/PI staining was also observed for MG63 cells after 3 days’ treatment with this fucoidan in the previous study, though there was no clear presence of a split population in the viable quadrant of the analysis [16].

The results also indicated that the early apoptotic population of the cells after *F. vesiculosus* crude fucoidan treatment was significantly larger than that of the cells treated with crude fucoidan from *S. filipendula*. Moreover, between the fractions as the MW increased, the late apoptosis cells also increased. These results indicate that fucoidan from *F. vesiculosus* caused apoptosis-like cell death and fucoidans from *S. filipendula*, especially at higher-molecular-weight fractions, led to cell necrosis.

### 2.6. Ultrastructure Examination of hES-MPs Treated with Fucoidan

To investigate the effects of different fucoidans on the ultrastructure of hES-MPs, TEM analysis was carried out. The results (Figure 6) showed that a typical hES-MP cell had a clear intricate tubular network of endoplasmic reticulum (ER) with inner diameter ranging from 0.25–0.5 μm in the perinuclear region in association with multivesicular bodies (MVBs) spread across the cytosol. The ER seemed to have an overall grey hue in the background with dark regions in the middle, which may be indicative of accumulation of materials. A common feature in most of the control cells was the presence a complex network of MVBs in the cytosol near the nucleus, ER or edge of the cells, which seemed to be autophagic precursors and autophagosomes. Additionally, the mitochondria were elongated with major axis length from 1–2 μm and had a clear cristae structure and pale granularity inside them. In the case of control MG63 cells reported before, no such extensive networks of MVBs were present in the cytosol.

When hES-MPs were treated with 10 μg/mL of crude fucoidan from *F. vesiculosus* or *S. filipendula*, it led to development of dense cytoplasmic lakes (more common with *S. filipendula* crude fucoidan), ranging in diameter from 0.5–3 μm, which seemed to be a result of accumulation of cellular material in the ER-like network. In the case of *F. vesiculosus* fucoidan, there were several condensed mitochondria with disintegrating cristae and dark appearance and, in some areas, mitochondria seemed to be breaking up into pieces smaller than 0.5 μm wide, surrounded by condensed cytoskeletal filaments, possibly actin, in the cytosol. In the case of hES-MPs treated with *S. filipendula* crude fucoidan, the mitochondria had dark membranes and distorted cristae. Lastly, some of the cells treated with *F. vesiculosus* fucoidan also had significantly enlarged autophagosomes or MVBs and less granulated cytoplasm, indicative of enhanced autophagocytosis and poorer cell health. Previously, dense mitochondria with distorted cristae, ER swelling and autophagosomes were also seen in MG63 cells treated with these crude fucoidans [16].

Human ES-MPs were also treated with 100 μg/mL of LMW, MMW and HMW fucoidans (Figure 7) and there was significantly more structural damage in cells treated with MMW and HMW compared to LMW. For instance, these cells had less dense cytoplasm and lost plasma membrane integrity, causing cell lysis or necrosis, which resulted in cellular fragments becoming detached from the main cell bodies. All fucoidans seem to have significantly enhanced autophagocytosis compared to control or either of the crude-fucoidan-treated cells, with a number of larger-sized autophagic precursors and autophagosomes in the cytosol. Their mitochondria were smaller in size and had dense membranes with some swelling or distorted cristae. In all conditions, several cells seemed to be undergoing necrosis-like death having condensation of chromatin material near the nuclear membrane, and, especially in the nucleoplasm of HMW treated cells, there were a number of dark circular regions that indicated high activity in the nucleolus, possibly RNA synthesis. Collectively, these results indicate that the fucoidan with higher molecular weight was more necrotic to cells and affected the ER, mitochondria, nuclear material and may have upregulated autophagocytosis in hES-MPs. These results are similar to those previously observed for MG63 cells under the treatment of these fucoidans.

Lysosomes are formed during autophagy for digestion of cellular materials. To confirm the above results, live hES-MPs after 5 days of fucoidan treatment were stained with Neutral Red. The results (Figure 8) showed that as the fucoidan dose increased, the intensity of staining also increased and verified that fucoidan indeed upregulated lysozyme formation and possibly autophagy in hES-MPs.

### 2.7. Effect of Fucoidan on Mitochondrial Health

To investigate if mitochondrial integrity was affected by fucoidan, live hES-MPs were stained with JC-1 after 3 days of fucoidan treatment. The results (Figure 9A) showed that for a dose of 0.5 μg/mL, there was the highest mean green to red ratio for crude fucoidan from *F. vesiculosus* and, for the 100 μg/mL dose, there was a slightly lower mean green to red ratio with significantly reduced mitochondrial staining for the same fucoidan (Figure 9B), which indicates mitochondrial degradation. On the other hand, all other conditions had a similar mean green to red ratio as the vehicle control. Mitochondrial degradation has also been observed for the MG63 cells after treatment with 100 μg/mL crude fucoidan from *F. vesiculosus*. However, while the HMW fraction of *S. filipendula* caused significantly high mitochondrial depolarisation in MG63 cells, it was not clearly visible for hES-MPs in this study.

Expression of cyt c in the hES-MPs after fucoidan treatment was also evaluated (Figure 9A,B). At 0.5 μg/mL, there were no significant differences between different conditions. However, at 100 μg/mL there was significantly more cyt c expression in the cytoplasm of cells treated with both crude fucoidans and the MMW fraction. There was the lowest cyt c expression seen in the case of LMW- and HMW-treated cells. These results indicate that fucoidan acts differently on mitochondria depending upon the species of brown algae and molecular weight fraction. For the MG63 cells reported before, enhanced cyt c expression was only visible after treatment with crude fucoidan from *F. vesiculosus*.

### 2.8. Localisation of Fucoidan in hES-MPs

To study the interaction of different fucoidans with hES-MPs cells, immunostaining for fucoidan was performed for cells after 3 days treatment (Figure 10). While the crude fucoidan from *F. vesiculosus* seemed clumped in the perinuclear region, the crude fucoidan from *S. filipendula* seemed to be evenly distributed in the entire cytosol of the cells. In the case of MG63 cells, however, penetration of crude fucoidan from *F. vesiculosus* seemed to be more limited as it formed clumps either near the edges of the cells or in the cytosol but the crude fucoidan from *S. filipendula* showed full penetration.

Human ES-MPs were also stained for fucoidan after treatment with different molecular weight fractions. The results (Figure 10) showed no significant staining for LMW, but visible penetration of MMW and HMW fractions, though their staining was not as intense as the crude fractions.

### 2.9. Effect of Fucoidan on Osteogenic Differentiation

To investigate early osteogenesis in hES-MPs, ALP activity assay was performed after 14 days of fucoidan treatment. The results (Figure 11A) showed no statistically significant differences between the crude fucoidans at any concentration. However, there was higher ALP activity/DNA content in 1 µg/mL MMW-treated cells compared to 1 µg/mL LMW- or HMW-treated cells.

When Alizarin Red S staining was performed on Day 28 of differentiation to assess mineral deposition, there was significantly more mineral deposited after treatment with 0.1 µg/mL of crude, LMW and HMW fucoidan from *S. filipendula* and then again for 0.2 µg/mL of all four fucoidans from *S. filipendula* (Figure 11B,C) compared to control. With the 1 µg/mL dose, there was more mineral per cell in the case of *S. filipendula* crude fucoidan compared to *F. vesiculosus* crude fucoidan and the least mineral in the case of HMW-treated cells compared to MMW- and LMW-treated cells.

Sirius Red assay was also performed on Day 28 of differentiation to assess collagen production. The results (Figure 11D,E) showed significantly reduced collagen after treatment with 0.1 to 1 µg/mL of crude fucoidan from *F. vesiculosus* but this negative effect was only visible at the 0.5 µg/mL dose for crude fucoidan from *S. filipendula*. Interestingly, at 0.2 µg/mL there was significantly higher collagen for MMW fucoidan compared to vehicle control but beyond 0.2 µg/mL there was a dose-dependent drop in values for MMW as well as LMW and HMW fucoidans, with the least collagen for HMW-treated cells. These results suggested that both crude fucoidans did not enhance late bone markers in hES-MPs but the one from *F. vesiculosus* and HMW from *S. filipendula* were detrimental at 1 µg/mL. However, doses of LMW and MMW from *S. filipendula* of 0.1–0.2 µg/mL may be more beneficial.

## 3. Discussion

In the last decade, a few studies have investigated the effects of fucoidans on osteogenic differentiation mostly using cancerous cell lines or murine osteoblasts [14,18,20,28,29,30] and less so using human osteoblasts or relevant stem cells [17,19,24]. This study therefore explores the effects of crude fucoidans derived from two different species of brown algae, namely, *F. vesiculosus*, which is the most widely reported fucoidan in the literature (and can be obtained commercially), and *S. filipendula* that grows abundantly on the Colombian coast and could provide a cost-effective source of fucoidan for medical use. The effect of different molecular weight fractions, 10–50 kDa (LMW), 50–100 kDa (MMW) and >100 kDa (HMW), of fucoidan from *S. filipendula* is also investigated. To our knowledge, this is the first time that the effects of *S. filipendula* fucoidans have been tested on human mesenchymal progenitor cells (hES-MPs) that represent healthy bone-forming cells. In this discussion, we will compare the effects on hES-MPs with the effects on osteosarcoma-derived MG63 cells using the same fucoidans reported in a previous study [16]. The aim was to understand if these fucoidan(s) could be used as an adjuvant drug for bone regenerative applications in osteosarcoma patients after resection for more efficient reconstruction.

FTIR-ATR analysis confirmed that all of the fucoidans had characteristic absorption bands of sulphated polysaccharides O-C-O, S=O and C-O-S. However, there was significantly higher sulphation levels and glycosidic bonds and lower uronic acid content in crude extract from *F. vesiculosus* compared to all other fucoidans, which may explain the differences in the mechanisms of action of this particular fucoidan. These results are a confirmation of our previous findings where weight percentages of glucose, fucose, sulphated sugars and uronic acid were measured [16,31]. A higher sulphation and lower uronic acid contents have been associated with high anticoagulant capacity [32], suggesting that the crude extract from *F. vesiculosus* may have clinical relevance as an anti-coagulant agent.

In terms of cell attachment, hES-MPs appeared to be more resilient than the MG63 cells towards fucoidans, which may be due to differences in cell size. This has implications for their use as an anti-osteosarcoma agent, where the aim would be to reduce the negative effects on healthy cells. While the hES-MP metabolic activity remained > 50% compared to control for all conditions at the highest fucoidan (100 µg/mL) dose even 48 h after seeding, the MG63 cells had < 50% metabolic activity 24 h after cell seeding in the presence of the same doses of the same fucoidans (crude, LMW and HMW fucoidans from *S. filipendula*). Moreover, the hES-MPs had visibly unaffected vinculin staining at the 0.5 µg/mL dose for all fucoidans but, in the case of the MG63 cells, negative effects of most fucoidans except the LMW fraction of *S. filipendula* were already visible at this dose. Proliferation of both cell types was inhibited in a dose-dependent manner, especially due to crude fucoidans or the higher-molecular-weight fucoidan. TEM, JC-1 staining and cyt c analyses confirmed that both cell types had mitochondrial damage, swollen ER and upregulated autophagy after fucoidan treatments, which validated that hES-MP cell death due to these fucoidans was stress induced. There was also consistency in the cell cycle analyses where both cell types underwent a G1 arrest with accumulation in the sub-G1 population after treatment with *F. vesiculosus* fucoidan but only a G1 arrest due to crude or higher-molecular-weight fucoidans from *S. filipendula*, which again validated that stress-induced apoptosis-like cell death was the main route of cell death due to *F. vesiculosus* fucoidan and stress-induced necrosis-like cell death was more prominent after treatment with the other fucoidans. Future studies exploring the effect of fucoidan on mitochondrial metabolism and mitochondrial DNA would be useful to support the above results related to mitochondrial damage.

Some studies have reported fucoidans increase ALP activity and mineralisation in human osteoblasts (10 µg/mL of fucoidan from unknown source, <30 kDa) [17], in human adipose-derived stem cells and human amniotic fluid stem cells (100 µg/mL fucoidan from *Uridina pinnatifida* with 25% organic sulphate and 60% polysaccharide content, >30 kDa) [24], in MSCs from human alveolar bone (0.1–1 µg/mL fucoidan from *Laminaria japonica* with 62% polysaccharides and 24% sulphate) [20] and also in human adipose-derived stromal cells (10 µg/mL fucoidan from *Ascophyllum nodosum*, 4.5 kDa) [19]. However, in this study no consistent effect on ALP activity was observed with any of the fucoidans between the doses of 0.1 to 10 µg/mL, and higher mineral or collagen formation was only seen at lower doses for *S. filipendula* fucoidans, i.e., from 0.1–0.2 µg/mL. This may be due to several reasons such as variation in different algal species, the methodology used to harvest fucoidans, the molecular weight used and also the varying sulphur contents in fucoidans, all of which can affect the bioactivity [33,34,35].

The higher sulphur content (33%) in *F. vesiculosus* fucoidan compared to all other *S. filipendula* fucoidans (15–25%) has been directly linked to a higher level of toxicity caused in MG63 cells [16]. It appears that this may also apply for hES-MPs where the sulphur content may regulate the osteoinductivity of these fucoidans. Higher sulphur content may cause a reduction in mineralisation and collagen formation in a dose-dependent manner, while fucoidans with lower sulphur contents such as LMW and MMW fractions from *S. filipendula* may be osteogenically stimulating, especially at lower doses (0.1–0.2 µg/mL), and support bone formation.

In this study, we also studied the penetration of fucoidans inside hES-MPs. Crude fucoidan from *F. vesiculosus* penetrated less easily into MG63 cells but entered more readily inside the hES-MPs, where sometimes it formed clumps in the perinuclear region. This may be attributed to the difference in the cell size (hES-MPs were almost three times the size of MG63 cells) and the amount of cytosol present. Moreover, Jang et al. [36] demonstrated that fucoidan-coated gold nanoparticles can be taken up by the cells through endocytosis and become entrapped in endosomes, which then migrate within the cytosol and aggregate in the perinuclear region, contributing to autophagy.

On the other hand, the crude fucoidan from *S. filipendula* seemed to penetrate more readily into both cell types as it was dispersed across the whole cytoplasm evenly. This suggests that fucoidans from different algal species penetrate differently inside in both cell types. Human ES-MPs were also stained for different molecular weight fractions of *S. filipendula* fucoidans and, interestingly, there was visible staining for MMW and HMW fractions inside the cells and not for the LMW fraction, which may be due to the specificity of the antibody itself. For example, Torode et al. [37] reported that the BAM3 epitope is most abundant in three Fucale species, *Pelvetia canaliculata*, *Fucus spiralis* and *Ascophyllum nodosum*, and least abundant in brown algae which are always submerged, such as *Sargassum muticum*.

Altogether, our results from the two studies suggest that higher doses of fucoidan, irrespective of the source, are toxic to cells (cancerous or healthy). However, if low doses of fucoidan could be delivered to the bone regenerative sites containing stem cells in a controlled manner over a long period of time, after an osteosarcoma resection (possibly in combination with a 3D scaffold), it could enhance osteogenesis in that region, with damaging effects on any remaining cancerous cells. The most potent fucoidans based on our current results were the LMW and MMW fractions of *S. filipendula* (in that order) that ranged from 10 to 100 kDa. However, future work is needed to test fucoidans in a co-culture system of hES-MPs and MG63 cells and also narrow down the molecular weight fraction which can accomplish the dual therapeutic purpose, while identifying the molecular mechanisms behind these effects.

## 4. Materials and Methods

All reagents were purchased from Sigma-Aldrich, Gillingham, UK unless otherwise stated. The plate reader used for absorbance and fluorescent readings was an Infinite F200 PRO TECAN microplate reader (Tecan, Theale, UK), unless otherwise stated.

### 4.1. Extraction and Purification of Fucoidan

Crude fucoidan from *F. vesiculosus* was bought commercially (F5631) and four types of fucoidans, i.e., crude extract, low-molecular-weight fraction (10–50 kDa, LMW), medium-molecular-weight fraction (50–100 kDa, MMW) and high-molecular-weight fraction (>100 kDa, HMW), were extracted from *S. filipendula* harvested from the coast of Colombia as described previously [16].

### 4.2. FTIR-ATR

The FTIR-ATR spectroscopy was performed in a Bruker Tensor II FT-IR with a Diamond ATR (Bruker Optics, Ettinger, Germany) and a liquid N_2_-cooled photovoltaic MCT detector. Briefly, 20 mg/mL samples at room temperature (RT) were used for the analysis and 64 scans with a resolution of 4 cm^−1^ against the background subtraction were assessed. The FTIR-ATR spectra were corrected for baseline and then analysed using OPUS 7.5 Software (Bruker Optics, Ettlingen, Germany).

### 4.3. Culturing of Human Embryonic Derived-Mesenchymal Progenitor (hES-MP) Cells

Human ES-MPs (passages 5 to 8, hES-MP002.5, Takara Bio Europe AB, Gothenburg, Sweden), which have been previously shown to mimic the behaviour of adult mesenchymal stem cells [38], were maintained in α-minimum essential medium (α-MEM) with Eagle’s salts (Corning, Flintshire, UK), 10% foetal bovine serum (FBS, Gibco, London, UK), 2 mM L-glutamine and 100 mg/mL penicillin–streptomycin at 37 °C and 5% CO_2_ in flasks coated with 0.1% type A gelatin from porcine skin prepared in deionised sterile water for 30 min. At 90% confluency, cells were trypsinised, collected and seeded at 10,000 cells/cm^2^ (unless otherwise mentioned) on plates or in flasks pre-coated with gelatin. To assess cell attachment, cells were seeded in the presence of different fucoidans and after 1.5 days, assessments were performed. To assess proliferation, treatment with fucoidan was started 48 h after cell seeding. For differentiation, cells were seeded at the same density and after 48 h standard culture medium was replaced with osteogenic medium containing 100 nM dexamethasone, 50 µg/mL ascorbic acid 2-phosphate and 0.5 mM beta-glycerophosphate in addition to ingredients mentioned above. Medium was replenished every 2–3 days.

### 4.4. Cell Metabolic Activity Assay

Cell metabolic activity was measured using PrestoBlue^®^ assay (ThermoFisher Scientific, Altrincham, UK). At each time point, the culture medium was removed, cells were washed with phosphate buffer saline (PBS) and then PrestoBlue^®^ working solution (PrestoBlue^®^ reagent mixed with pre-warmed Hanks balanced salt solution at a ratio of 1:9 (*v*/*v*)) was added [16]. Three blank wells containing only PrestoBlue^®^ working solution were also included. During the kinetic phase of reactions, after gentle mixing of cells, 100 μL aliquots per well were taken and the fluorescence intensity was measured using 530 nm excitation and 590 nm emission filters on the microplate reader. Cell metabolic activity was expressed after removing blank values and normalising to the control (0 μg/mL fucoidan dose) for respective fucoidans at each time point.

### 4.5. Focal Adhesion Staining

The culture medium was removed and after washing with PBS, the cells were fixed with 3.7% paraformaldehyde in PBS for 15 min at RT. Staining was performed using the Actin Cytoskeleton and Focal Adhesion Staining Kit (Millipore, Feltham, UK) according to the manufacturer’s instructions [16]. Cells were washed twice with wash buffer, permeabilised with 0.1% Triton X-100 in Dulbecco’s phosphate buffer saline (DPBS) and then again washed with wash buffer. The cells were blocked using a blocking solution before adding anti-vinculin antibody for 1 h at RT. Then, cells were washed twice using wash buffer before incubating in Gt X Ms, FITC-conjugated secondary antibody (AP124F, Millipore, Feltham, UK) and TRITC-labelled phalloidin for 1 h at RT. Finally, cells were washed with wash buffer and stained with DAPI (4′,6-diamidino-2-phenylindole) before being visualised using a Nikon A1 confocal microscope (Nikon Instruments Inc., New York, NY, USA) with a 60× oil lens.

### 4.6. DNA Content Assay

The culture medium was removed, cells were washed with PBS and sterile deionised water was added for lysing cells through three freeze–thaw cycles at −80 °C. One hundred microlitres of test lysate was used and DNA content was quantified using a Quant-i™ PicoGreen^®^ dsDNA Assay kit (ThermoFisher Scientific, UK) as per the manufacturer’s instruction [39]. One hundred microlitres of PicoGreen working solution was added to each test well and three blank wells. After a 10 min incubation period at RT, fluorescence intensity readings were measured on a microplate reader using 480 nm excitation and 520 nm emission filters. DNA concentration standard curve was also prepared to extrapolate the measured values. DNA content was expressed after subtracting the reading for blank wells and normalising to the control (0 μg/mL fucoidan dose) for respective fucoidans.

### 4.7. Giemsa Staining

After removing culture medium, cells were fixed with methanol and washed with deionised water before immersing them in Giemsa stain for 15 min at RT. After another wash with deionised water, cells were examined under a MOTIC^®^ AE 2000 inverted microscope, Kowloon, Hong Kong.

### 4.8. Cell Cycle Analysis

Two days after cell seeding, culture medium was changed to α-MEM without serum for 16 h to synchronise the cell cycle before starting fucoidan treatment. After 3 days, medium was removed and cells were washed with DPBS, before adding 70% ethanol and vortexing immediately. After another DPBS wash, propidium iodide (PI) working solution and RNase working solutions were added and incubated at 37 °C for 1 h. The samples were analysed using a BD LSRII flow cytometer and FlowJo v10 software (BD, Wokingham, UK).

### 4.9. Annexin V/PI Staining

Cells were seeded and, after 48 h, treatment with fucoidan was started. After 5 days, the staining was performed using a TACS^®^ Annexin V-FITC Kit (Trevigen, Bio-Techne, Minneapolis, MN, USA). The cells floating in culture medium and attached to the flasks were collected, washed with DPBS and stained with Annexin V and PI for 15 min at RT [16]. After washing the cells with binding buffer, samples were assessed in a BD LSRII flow cytometer and data were analysed using FlowJo v10 software. For PI only and Annexin V only controls, cells were cultured with 1% saponin and 30% H_2_O_2_ solution, respectively. Autofluorescence signal was excluded by including cells without any stain.

### 4.10. TEM

For TEM, the cells were seeded and, after 48 h fucoidan treatment was started. Three days later, cells were collected and prepared for TEM as described before [16]. Electron micrographs were recorded using an Orius 1000B Gatan digital camera (AMETEK GmbH, Unterschleissheim, Germany) and Gatan Microscopy Suite^®^ version 3.x (AMETEK GmbH, Unterschleissheim, Germany).

### 4.11. Neutral Red Staining

The stock solution of Neutral Red was prepared by adding 0.2 g of Neutral Red (NR, Abcam, Cambridge, UK) to 50 mL of sterile deionised water. For working solution, stock solution was diluted 1 in 80 with α-MEM and syringe filtered using a 0.2 μm filter [40]. For cell viability measurements, the culture medium was removed, cells were washed with PBS, then NR/α-MEM solution was added to each well along with three blank wells. After 3 h incubation at 37 °C, 5% CO_2_, cells were washed with PBS and imaged using a Nikon Eclipse TS100 (Nikon Instruments Inc., New York, NY, USA).

### 4.12. JC-1 Staining

Human ES-MP cells were seeded and left overnight to attach. Next day, treatment with fucoidan was started and after 3 days culture medium was removed and cells were washed with DPBS before incubating them in 10 µM of JC-1 dye (5,50,6,60-tetrachloro-1,10,3,30-tetraethylbenzimi-dazolylcarbocyanine iodide, T3168, ThermoFisher Scientific, UK) [41] in culture medium for 30 min [16]. Cells were then washed with DPBS and imaged under a ZEISS LSM 880 with Airyscan with a Plan-Apochromat 40×/1.3 oil lens using excitation of 488 nm and emission of 496–550 nm (for green monomers) and 576–719 nm (for red aggregates).

### 4.13. ALP Activity Assay

The ALP activity was measured using p-Nitrophenyl Phosphate (pNPP) Phosphatase Substrate (ThermoFisher Scientific, UK). Briefly, 100 μL of cell lysates and 100 μL of substrate were added to each assay well and after 30 min of reaction absorbance at 405 nm was measured on the microplate reader. The ALP activity per cell was expressed as mean pNP/μg of DNA normalised to respective vehicle controls.

### 4.14. Sirius Red Assay

A working solution of 0.1% (*w*/*v*) Sirius Red was prepared in saturated picric acid. The culture medium was removed, cells were washed with PBS and fixed with 3.7% paraformaldehyde in PBS. After another PBS wash, the Sirius Red working solution was added for 30 min [39]. Then, cells were washed with deionised water. For quantification, a destaining solution (1:1 0.1N NaOH:methanol) was added for 30 min. Three 100 μL extracts per well were transferred to a clear 96-well plate and absorbance readings were taken on a microplate reader at 540 nm. Readings were normalised to DNA content and respective vehicle controls (0 μg/mL).

### 4.15. Alizarin Red S Staining

The culture medium was removed, cells were washed once with PBS and fixed with 3.7% paraformaldehyde in PBS. After another PBS wash, 1% (*w/v*) solution of Alizarin Red S was added for 30 min [39]. Cells were then washed with deionised water. For quantification, a destaining solution (5% perchloric acid in deionised water) was added for 30 min. Three 100 μL extracts per well were transferred to a clear 96-well plate and absorbance readings were taken on a microplate reader at 405 nm. Readings were normalised to DNA content and respective vehicle controls (0 μg/mL).

### 4.16. Immunostaining for Cytochrome c (cyt c) and Fucoidan

The culture medium was removed, cells were washed with PBS and fixed with 3.7% paraformaldehyde in PBS. The samples were permeabilised with 0.1% Triton X-100 in DPBS, then washed with DPBS before blocking with 3% goat serum in 1% BSA in DPBS. Then, samples were incubated overnight at 4 °C with BAM 3 (Hybridoma extracts containing anti-fucoidan primary antibody, 1:10 dilution) [37] or 1 µg/mL of anti-cyt c primary antibody (ab13575, Abcam, UK). The next day, samples were washed twice in DPBS and incubated with 1 µg/mL of Alexa Fluor 674 (red) goat anti-rat IgG pre-absorbed (ab150167, Abcam, UK) or Gt X Ms, FITC-conjugated secondary antibody (AP124F, Millipore, 1:100 dilution) in blocking solution for 2 h at RT. After again washing with DPBS, cells were stained with Acti-stain 488 phalloidin (PHDG1-A, Cytoskeleton, Inc., Denver, CO, USA) if needed and 10 μg/mL DAPI in 1% BSA in DPBS. Finally, cells were visualised using a Nikon A1 confocal microscope with a 60× oil lens. Image analysis was performed using ImageJ Version 2.0.0-rc-69/1.52p [40].

### 4.17. Statistical Analysis

All statistical analyses were performed using IBM SPSS Statistics 22. The mean and standard deviation values were estimated for at least three replicate samples in all experiments, except JC-1-stained images where 2 fields of view equating to at least 10–15 cells per condition were analysed. For comparisons, one-way or two-way ANOVA was performed. Fucoidan type and concentration were two fixed factors. For pairwise comparisons, post hoc analyses using least significant difference (LSD) were carried out. *p* values < 0.05 were considered significant. # indicates relative to respective vehicle controls (0 µg/mL) and * indicates *p* < 0.05 for pairwise comparisons between groups shown.

## Figures and Tables

**Figure 1 ijms-25-02085-f001:**
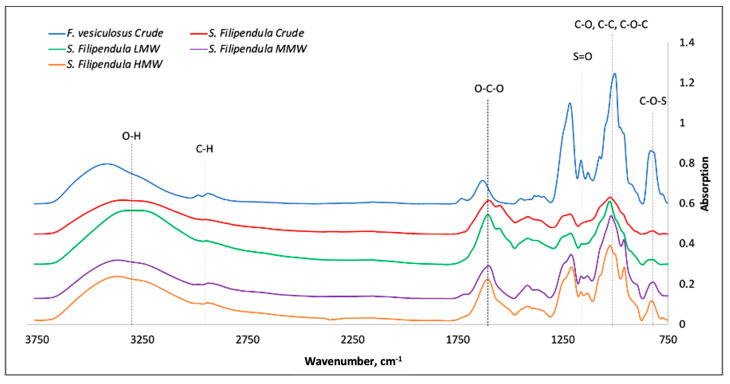
FTIR-ATR spectra of fucoidans and the corresponding peaks.

**Figure 2 ijms-25-02085-f002:**
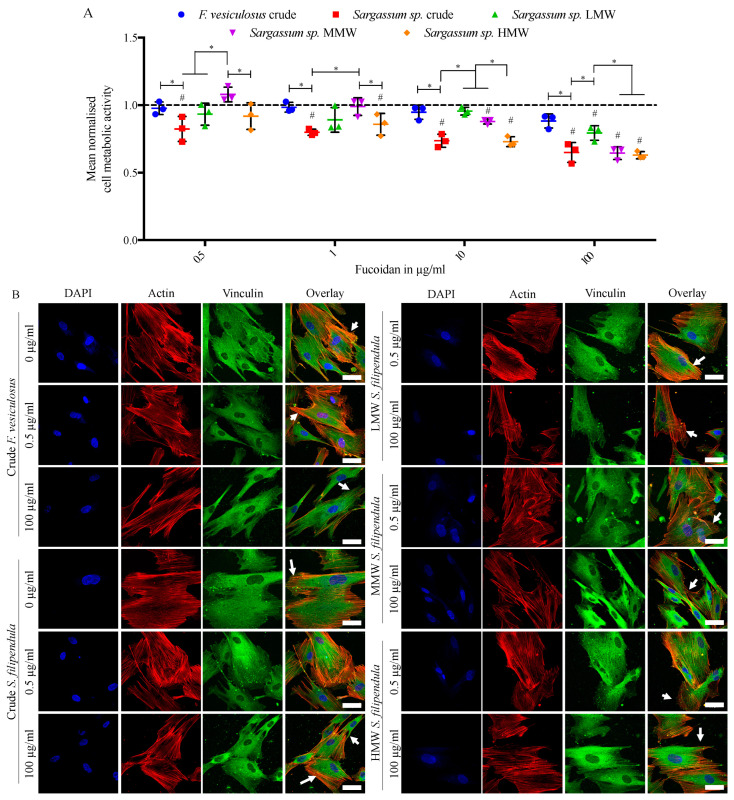
Effect of different fucoidans on hES-MP attachment. Mean ± S.D. of (**A**) cell metabolic activity (n = 3), - - - - indicates vehicle control (0 µg/mL). (**B**) Max intensity z-projections for DAPI (blue), actin (Texas Red) and vinculin (FITC/green) stained cells with overlays, 48 h after seeding in presence of fucoidan. Scale bars—50 µm, arrows—reduced vinculin expression, * *p* < 0.05, # *p* < 0.05 relative to respective vehicle control.

**Figure 3 ijms-25-02085-f003:**
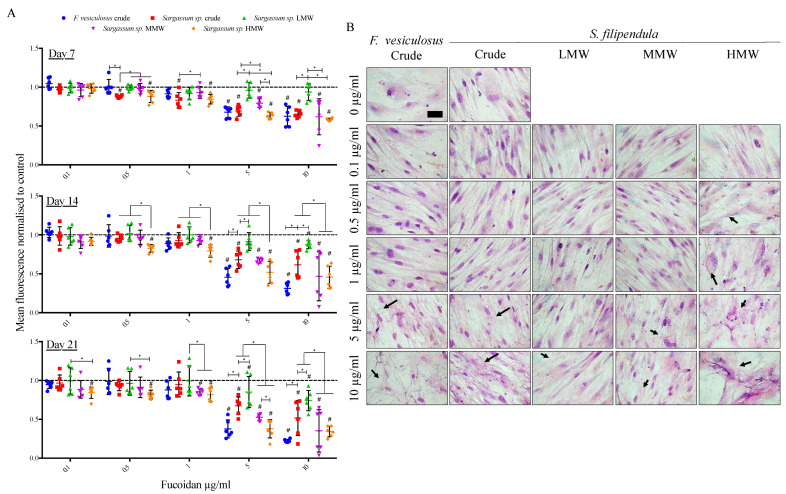
Mean ± S.D. of (**A**) cell metabolic activity (n = 6) of hES-MPs after 7, 14 and 21 days of treatment with different fucoidans from 0–10 μg/mL, - - - - indicates vehicle control (0 µg/mL). * *p* < 0.05, # *p* < 0.05 relative to respective vehicle control. (**B**) Giemsa-stained hES-MPs after 21 days’ treatment. Arrows—distorted cell structure. Scale bar—200 μm.

**Figure 4 ijms-25-02085-f004:**
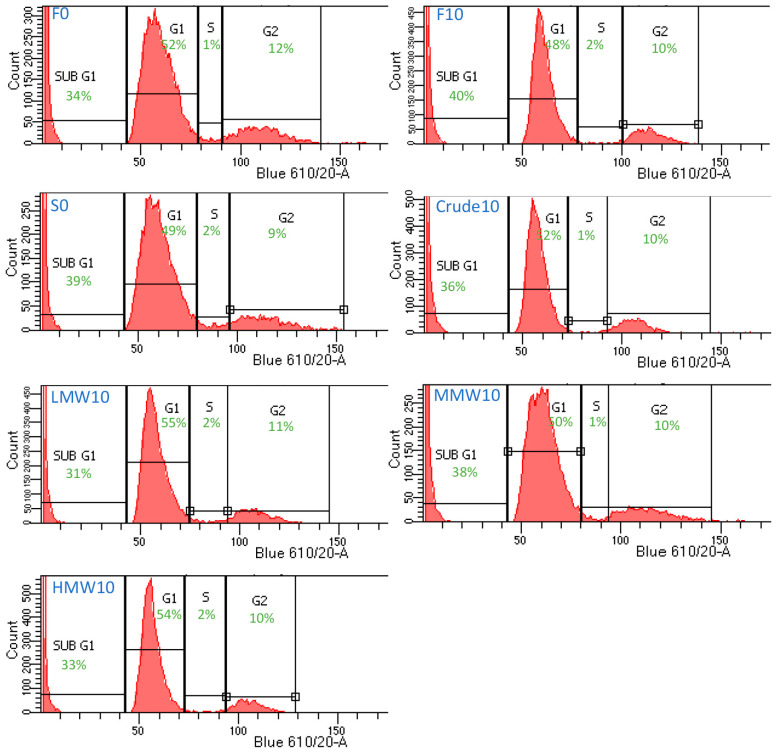
Cell cycle analysis of hES-MPs after 5 days’ treatment with different fucoidans at 0 and 10 μg/mL doses. Nearly 10,000 single cell events were measured after staining with PI (n = 1) and populations were split into sub-G1, G1, S and G2 phases.

**Figure 5 ijms-25-02085-f005:**
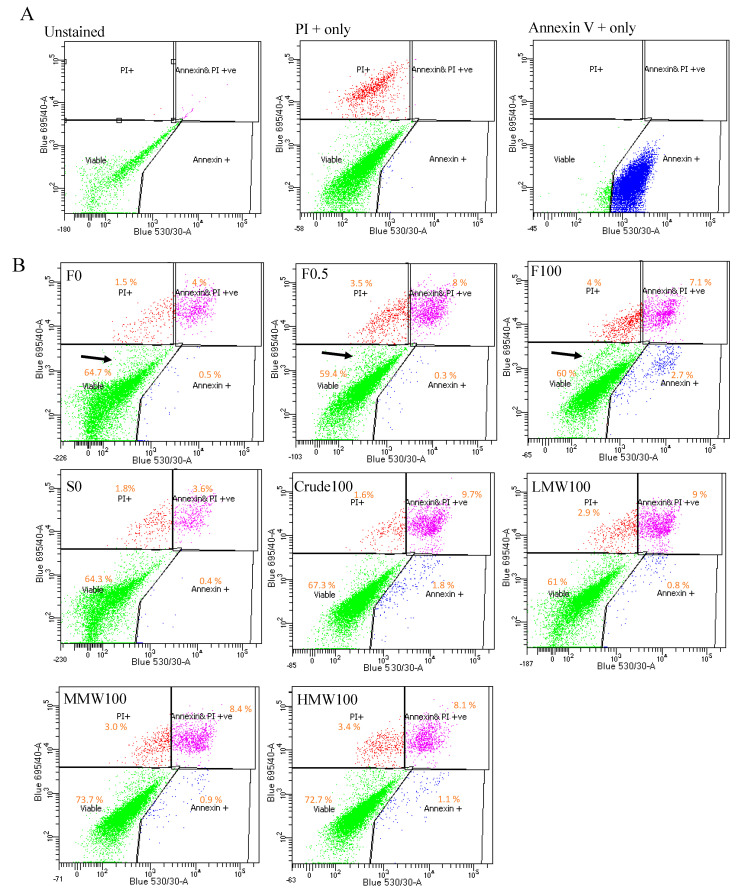
Apoptotic analysis for hES-MPs using Annexin V/PI staining after 5 days’ treatment with different fucoidans. (**A**) Gating strategy using unstained, PI+ only and Annexin+ only cells. (**B**) Cells were seeded at 10,000 cells/cm^2^ and fucoidan treatment was started next day. Representative gated channels for percentage of viable, Annexin V+ and PI+ cells (late apoptotic), Annexin V+ (early apoptotic) cells and PI+ (dead) cells are shown. Black arrows indicate a separate set of populations in the viable gate after *F. vesiculosus* fucoidan treatment.

**Figure 6 ijms-25-02085-f006:**
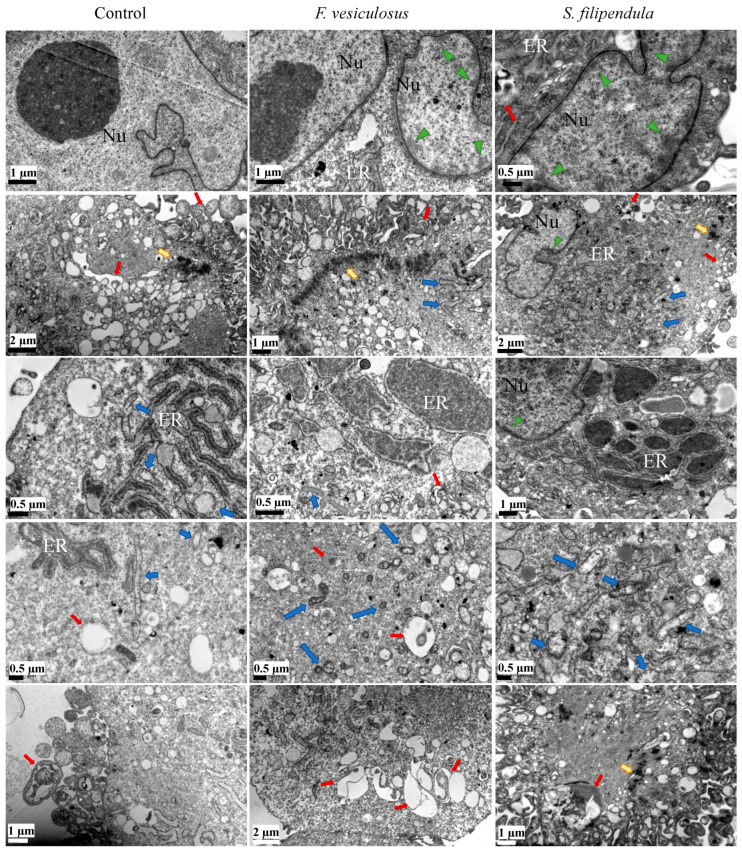
TEM of hES-MPs cells treated with 10 μg/mL of crude fucoidan from *F. vesiculosus* or *S. filipendula* for 3 days (at least 10–15 cells analysed per condition). Nu—nucleus, green arrow heads—condensation of chromatin material, red arrows—autophagosomes, yellow arrows—dark regions of actin-like filament condensation, ER—endoplasmic reticulum network, blue arrows—mitochondrion.

**Figure 7 ijms-25-02085-f007:**
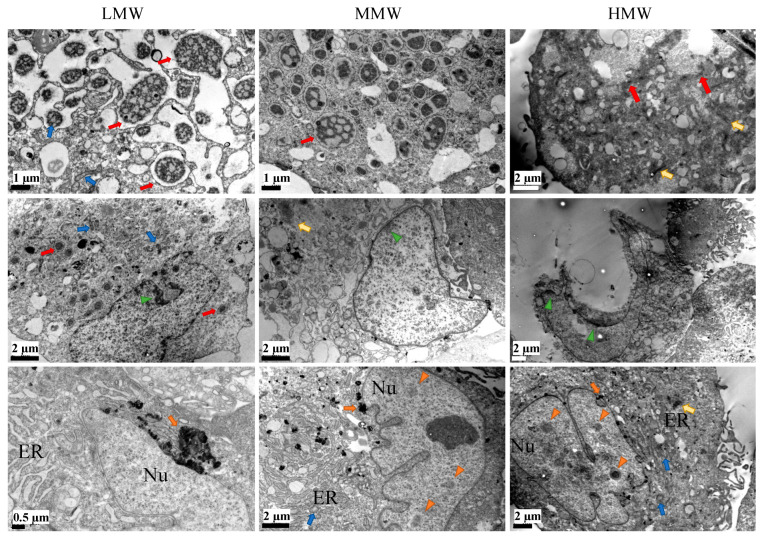
TEM of hES-MP cells treated with 100 μg/mL of LMW, MMW and HMW fractions of fucoidan from *S. filipendula* for 3 days (at least 10 cells were analysed per condition). Red arrows—autophagosomes, blue arrows—mitochondrion, green arrow heads—condensation of chromatin material, orange arrows—dark deposits, orange arrow heads—dark circular spots in the nucleus (regions of high RNA synthesis), yellow arrows—dark regions of actin-like filament condensation.

**Figure 8 ijms-25-02085-f008:**
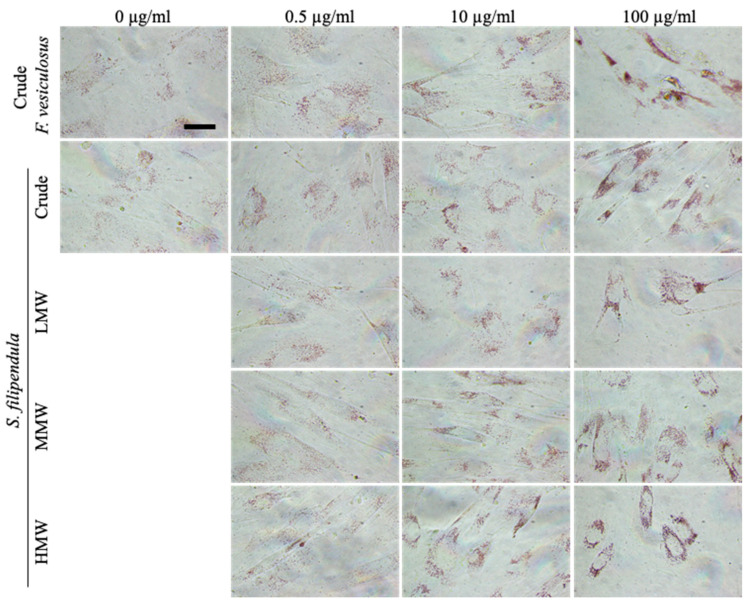
Neutral-Red-stained hES-MPs after 5 days’ treatment with different fucoidans at different concentrations. Notice the increase in intensity of Neutral Red staining (positive for lysosomes) as the fucoidan dose increases. Scale bar—200 μm.

**Figure 9 ijms-25-02085-f009:**
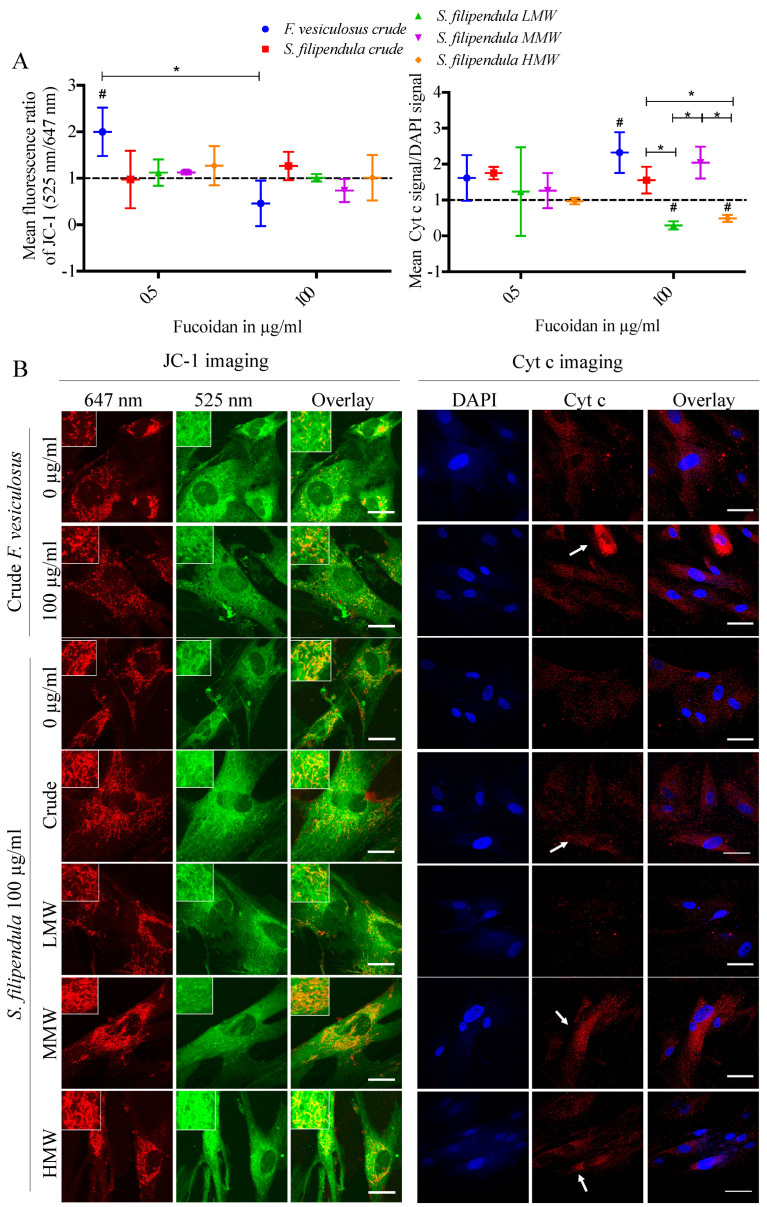
Assessment of mitochondria in hES-MP cells after 3 days of treatment with different fucoidans. (**A**) Mitochondrial membrane potential changes measured using JC-1 staining (at least 3 cells were analysed per condition) and quantification of cyt c signal to DAPI signal (n = 3 fields of view). (**B**) On the left, representative images of live cells taken under 525 nm (green) and 647 nm (red) channels with overlays after JC-1 staining. Scale bars—25 µm. The indented sections—84 µm × 84 µm. On the right, representative overlay images of cells stained with cyt c (red, white arrows) and DAPI (blue). Scale bar—50 µm. * *p* < 0.05, # *p* < 0.05 relative to respective vehicle control.

**Figure 10 ijms-25-02085-f010:**
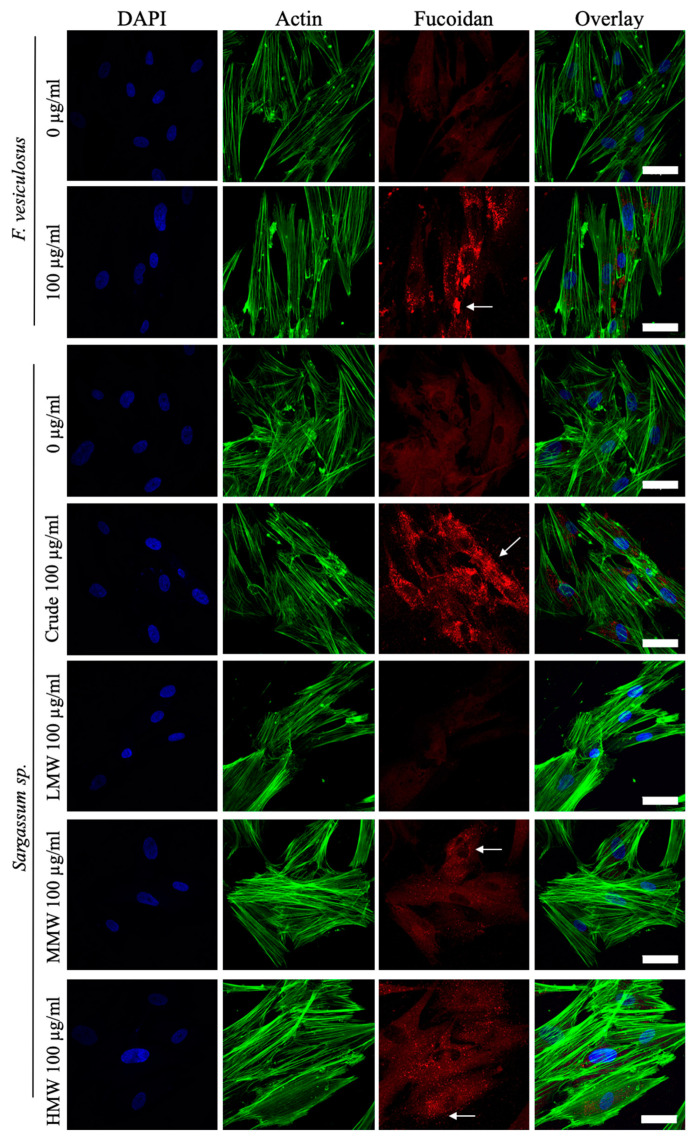
Penetration of different fucoidans in hES-MPs after 3 days of treatment. Max intensity z-projections for DAPI (blue), fucoidan (red) and actin (FITC/green) stained cells with overlay images are shown. At least 3 fields of view were analysed per condition. Notice the difference in patterns of staining for different crude fucoidan types (arrows). Scale bars—50 µm.

**Figure 11 ijms-25-02085-f011:**
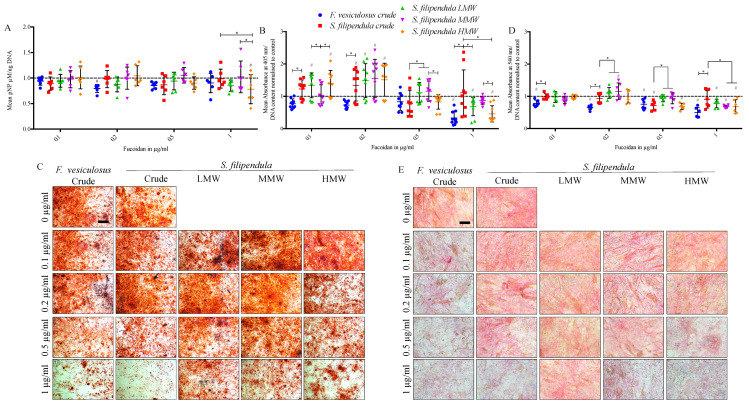
Effect of different fucoidans on osteogenic potential of hES-MPs. (**A**) ALP activity on Day 14 (n ≥ 6), (**B**,**C**) Alizarin Red staining for mineralisation on Day 28 (n ≥ 9) and (**D**,**E**) Sirius Red staining for collagen estimation on Day 28 (n ≥ 9). - - - - indicates control (0 µg/mL). Scale bar—200 µm (**C**), 100 µm (**E**). * *p* < 0.05, # *p* < 0.05 relative to respective vehicle control.

**Table 1 ijms-25-02085-t001:** The functional groups in fucoidans based on FTIR-ATR spectra.

Wavelength, cm^−1^	Functional Group Assignment
3400–1600	O-H, C-H, O-C-O; C-OH
1610–1430	Uronic acid
1240–1210	Sulphated group extension
900–950	Glycosidic bonds
810–840	Sulphated group

## Data Availability

Raw data can be made available upon reasonable request.

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
