# Peer review of "The Effects of Fucoidan Derived from Sargassum filipendula and Fucus vesiculosus on the Survival and Mineralisation of Osteogenic Progenitors"

_ijms, 2024, doi:10.3390/ijms25042085_

Round 1
Reviewer 1 Report
Comments and Suggestions for Authors
The authors studied fucoidan's potential for treating osteosarcoma by testing its effects on human embryonic stem cell-derived mesenchymal stem cells. They aimed to determine if fucoidan has anti-cancer and bone regeneration properties for potential osteosarcoma treatment. In general, the manuscript is well-crafted and easily understandable. I trust that my suggestions may enhance the translational significance and clinical applicability of the authors' compelling discoveries.
Here are my suggestions.
1. The study shows promise for fucoidan as an adjuvant therapeutic agent, but additional molecular profiling is needed to determine efficacy and safety. Whole genome/exome sequencing of treated cell populations could identify mutations indicating potentially carcinogenic effects.
2. Testing fucoidan in combination with standard chemotherapeutic agents could provide insights into pharmacodynamic interactions and support its use as an adjunct therapy.
3. In Figures 2A, 3A, 9A, and 11A, the authors employed various metrics like cell metabolic activity and ALP activity to assess the impact of different fucoidans. Given the use of multiple statistical tests, it is expected that corrections were applied such as FDR.
Author Response
The response is attached as a word document

Reviewer 2 Report
Comments and Suggestions for Authors
Dear authors, i present a review of the manuscript «The effects of fucoidan derived from Sargassum filipendula and Fucus vesiculosus on survival and mineralisation of osteogenic progenitors».
Short summary:
This work meets the requirements of the journal and is devoted to the study of the total fractions of fucoidans with different molecular weights isolated from Sargassum filipendula and Fucus vesiculosus on the survival and mineralization of osteogenesis precursors. This study contributes to the study of the pharmacological properties of fucoidans. Expands the possibilities of their use in bone cancers.
Specific comments:
2. Results
2.1. Characterisation of fucoidans
For ease of perception of the material and interpretation of the data, it is necessary to place Figure 1 (IR spectra) in front of Table 1.
It is necessary to specify the chemical composition of the studied fucoidans: for example, the molar ratio of glucose and fucose in all three fractions, it is also necessary to specify how many components each fraction consists of, indicating the molecular weights of each substance and the percentage in the mixture.
It is necessary to show the purity of the studied fucoidans from phenolic compounds and proteins (UV spectrum and qualitative reaction to protein, amino acids).
It is also necessary to specify a method for determining the molecular weight of the polysaccharides under study.

Author Response

(The authors gave the same response as above.)

Reviewer 3 Report
Comments and Suggestions for Authors
This study stated the effects of fucoidan derived from Sargassum filipendula and fucus vesiculosus on survival and mineralization of osteogenic progenitors. Results revealed that LMW and MMW doses of < 200 ng/ml were least toxic and showed potential osteoinductivity. The study inderscores multifaceted impact of fucoidans on osteoprogenitor cell and highlights the delicate balance between potential therapeutic benefits and the challenges involved in using fucoidan for post-surgery treatments in osteosarcoma patients. This study is significant for the therapy of osteosarcoma patients. However, the further modification was needed to do for introduction and discussion.
Main comments:
1 Why author did these studies is not clearly state in the introduction.
2 The discussion should be performed around your main findings.
Minor comments:
1 In abstract, the abbreviations of FTIR-ATR, TEM, and JC-1 were first time appearance in the main text that should use the whole name.
2 Line 46, add a comma after gonadal dysfunction.
3 line 51, add a comma after cadaveric grafts.
4 line 54, add a comma after non-union.
5 line 62, add a comma after children.
6 The abbreviations of ALP, OCN, JNK, and BMP were first time appearance in the main text that should use the whole name.
7 line 62, add a comma after cell death.
Comments on the Quality of English LanguageModerate editing of English language required
Author Response

(The authors gave the same response as above.)

Round 2
Reviewer 1 Report
Comments and Suggestions for Authors
The authors have made their best efforts to address my comments from the previous round of review. I still have one suggestion. 1) The font is quite blurred in Figure 1, could you improve the clarity of the font in Figure 1?